# Noma (cancrum oris): A scoping literature review of a neglected disease (1843 to 2021)

**Elise Farley**[1,2]*, **Ushma Mehta**[3], **M. Leila Srour**[4], **Annick Lenglet**[5,6]

**1** Noma Children's Hospital, Médecins Sans Frontières, Sokoto, Nigeria, **2** Nudibrink Research Consultancy, Cape Town, South Africa, **3** Centre for Infectious Disease Epidemiology and Research, University of Cape Town, Cape Town, Western Cape, South Africa, **4** Health Frontiers, Vientiane, Laos, **5** Médecins Sans Frontières, Amsterdam, The Netherlands, **6** Department of Medical Microbiology, Radboudumc, Nijmegen, The Netherlands

* elisefarley@gmail.com

## Abstract

### Background

Noma (cancrum oris) is an ancient but neglected and poorly understood preventable disease, afflicting the most disenfranchised populations in the world. It is a devastating and often fatal condition that requires urgent and intensive clinical and surgical care, often difficult to access as most cases of noma occur in resource-limited settings. We conducted a scoping review of the literature published on noma to understand the size and scope of available research on the disease and identify research gaps that need to be addressed to evolve our understanding of how to address this disease.

### Methods

We searched 11 databases and collected primary peer reviewed articles on noma in all languages, the final search was conducted on 24th August 2021. The oldest manuscript identified was from 28th March 1843 and the most recently published manuscript was from 3rd June 2021. Search terms included cancrum oris and noma. Data was extracted using a standardised data extraction tool and key areas of interest were identified. The Preferred Reporting Items for Systemic review and Meta-Analyses requirements were followed.

### Results

The review included 147 articles, the majority of the studies (n = 94, 64%) were case reports. Most manuscripts (n = 81, 55%) were published in the 2000s, 49 (33%) were from the 1900s and 17 (12%) from the 1800s. The main areas of interest identified were the history and epidemiology of the disease, noma's clinical progression and aetiology, treatment regimens, mortality rates and the risk factors for the development of noma.

### Conclusions

Noma has been reported in the literature for hundreds of years; however important gaps in our understanding of the disease remain. Future research should focus on determining the

**Data Availability Statement:** All relevant data are within the manuscript and its Supporting Information files.

**Funding:** The author(s) received no specific funding for this work.

**Competing interests:** The authors have declared that no competing interests exist.

burden and distribution of disease; the true mortality rate, pathogenic cause(s) and the factors that influence prognosis and outcomes after treatment.

## Author summary

Noma is a devastating and often fatal condition that mainly affects children in severely disenfranchised communities. Noma is preventable and requires urgent basic medical care in the early stages of disease. Once the disease reaches the last stage, sequelae, survivors require expert surgical care, usually difficult to access as most cases of noma occur in resource-limited settings. We conducted a scoping review of the literature published on noma to understand the size and scope of available research on the disease and to identify research priorities that will evolve our understanding of how to eradicate this disease. Our review showed that noma has been reported in the literature for hundreds of years; however several major gaps in knowledge still exist. There is appreciation among the small community of clinicians and researchers involved in noma care and research that these gaps in knowledge impact on the ability to develop and implement sound evidence-based policies and activities aimed at eradicating noma from communities that continue to be afflicted by this ancient disease. The main focus of future research should be to study the burden and distribution of disease; the true mortality rate, and the pathogenic cause(s) and the factors that influence prognosis and outcomes after treatment.

## Background

Noma is a rapidly progressing infection of the oral cavity, associated with a reported 90% mortality rate within weeks after onset, if left untreated [1]. Noma mostly affects disenfranchised children who lack access to basic nutrition, hygiene services, and health care, although cases are reported in immunocompromised adults [1]. The pathogenesis of noma is poorly understood [1]. Commonly available broad-spectrum antibiotics can be used to treat the early reversible stages of noma [1]. Once noma progresses past these stages, the sequelae of noma are numerous and include difficulty in eating, drinking, seeing and breathing[1,2]. For those who seek care for these sequelae, it can mean hospital stays of many months with multistage surgical treatments that can take years to complete. Therefore, noma is associated with a high degree of morbidity for survivors, and this often has a significant impact on family members given the long-lasting and often permanent sequelae. Noma is an important public health issue and its existence is a painful reminder of the existing global inequalities in food distribution, health care access and living conditions. We conducted a scoping review of the literature on noma to consolidate the information available and to understand the size and scope of available research on this disease.

## Methods

This scoping review was conducted in line with Preferred Reporting Items for Systemic review and Meta-Analyses (PRISMA) requirements (S1 PRISMA checklist) [3].

### Databases searched

The following databases were searched manually for articles to include in this literature review: PubMed; PsycINFO via Ebsco Host; Science Direct; Social Science Citation Index via Web of

Science; MEDLINE via PubMed; Cumulative Index to Nursing and Allied Health Literature via Ebsco Host; Cochrane Library; Population Information Online; LILACS; SciELO and Scopus. The final search was conducted on 24 August 2021.

### Searching methods utilized

An initial search of each database was done online. All articles identified were listed. Available full text articles were downloaded. Outstanding articles were sourced through the University of Cape Town library or via the corresponding authors.

### Eligibility criteria

The databases and journals were searched using the following eligibility criteria: 1) noma, cancrum oris related; 2) peer reviewed; 3) primary study (a study including primary data collection, literature reviews and opinion pieces were not included); 4) addressed a main area of interest; 5) any publication date; 6) all study designs; and 7) all languages.

### Search terms

Databases were searched with the following terms: "cancrum oris" OR "noma" OR "cancrum oris cases" OR "cancrum oris defects" OR "cancrum oris like lesions" OR "cancrum oris noma" OR "cancrum oris, noma".

### Data extraction

Data from the eligible studies were extracted using a standardized data extraction tool. Data extracted included: title; author; journal; year of publication; geographic location of first author; geographic location of study; number of cases/individuals in study; research question/aim; methodology; analysis; results; area of interest; conclusions; implications for future research and practice; gaps in knowledge and any other noteworthy comments.

Our main areas of interest were the history and epidemiology of the disease, noma's clinical progression and aetiology, treatment regimens, mortality rates and the risk factors for the development and progression of noma.

All non-English papers were translated into English using Google translate, anything unclear was checked with a native language speaker.

### Analysis

A manual analysis was conducted by grouping individual factors within the areas of interest (the history and epidemiology of the disease, noma's clinical progression and aetiology, treatment regimens, mortality rates and the risk factors for the development and progression of noma). These areas of interest were explored in-depth and findings on each area are concisely reported.

## Results

Our initial search identified 200 full text articles for review, of which 147 were included in the literature review (full list of included manuscripts attached in S1 Data). The oldest manuscript identified was from 28[th] March 1843, and the most recently published manuscript was from 3[rd] June 2021. Most manuscripts (n = 81, 55%) were published in the 2000s, 49 (33%) were from the 1900s and 17 (12%) were from the 1800s. Fifty three manuscripts were excluded as they either did not directly relate to noma, did not meet the inclusion criteria, or they were not considered to be primary research. The majority of the studies (n = 94, 64%) were case reports (Fig 1).

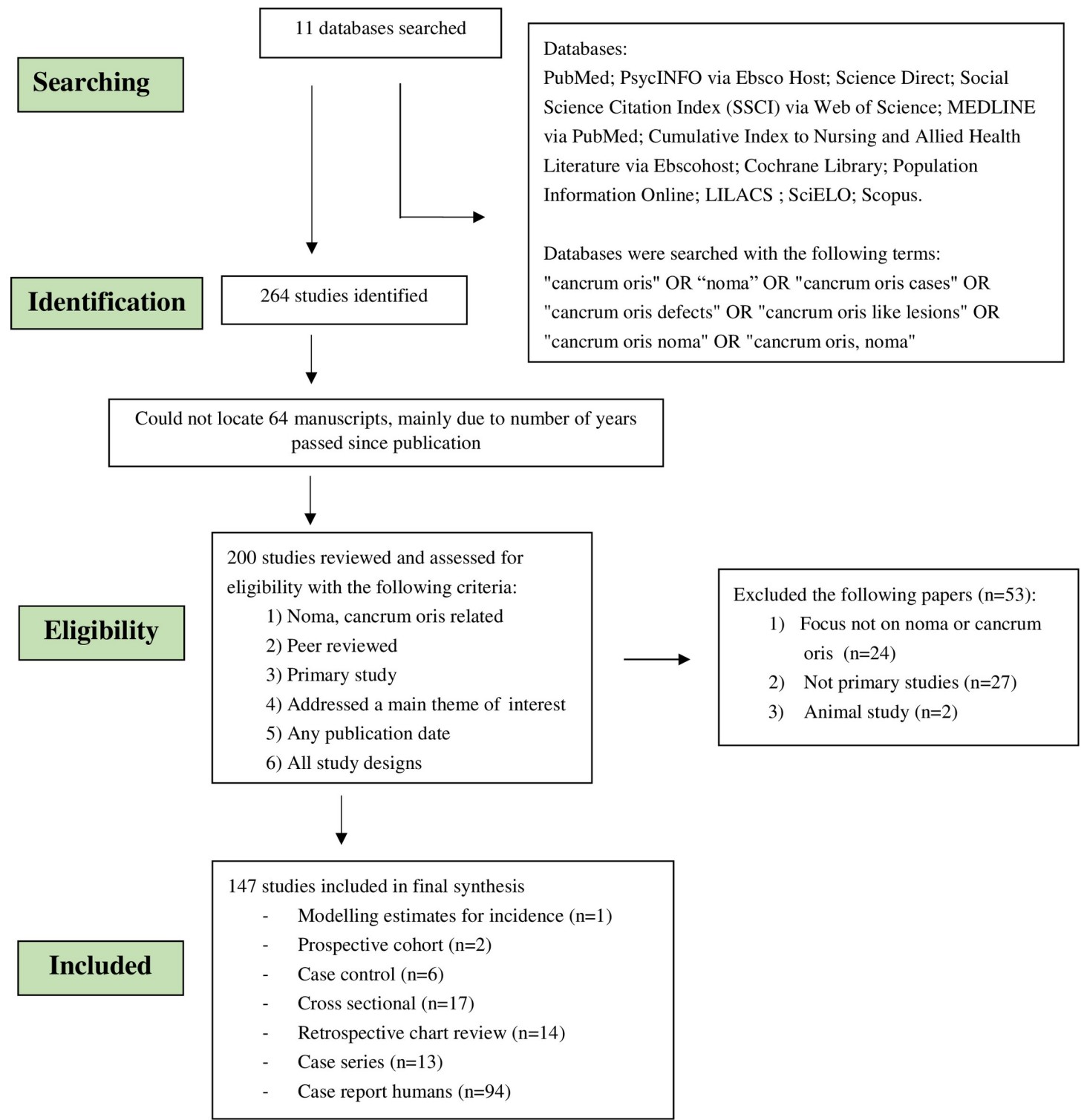

**Fig 1. Flow diagram of databases searched and articles included in the noma scoping review.**

## History of noma and names for the disease

The word 'noma' is derived from a Greek word which, loosely translated, means 'to devour' [4]. It was first used by Dutch surgeon Cornelis van de Voorde in 1680, for a rapidly-spreading ulceration originating in wet soft tissues 'typical of the mouth' [5]. The term 'noma' and 'cancrum oris' are currently used interchangeably [6–9]. In the 1800's, there were ongoing discussions about the usage of the terms with some viewing them as two separate diseases [10]. In a publication from 1862, 'noma' referred to ulcerative stomatitis (lesions on the skin or on the internal mucosal surface of the mouth) and 'cancrum oris' referred to gangrenous stomatitis (death of the tissue of the mouth) [10]. There are very early reports of clinical conditions similar to noma by physicians such as Hippocrates (460–370 BC) and Galen (129–200 AD) [11]. However; it was subsequently reported that this referred to general ulceration of the body and not noma as the disease is currently understood [12]. The first clinical description of the disease we now call noma, was written by Battus in 1620, who labelled it 'water canker' [5,12]. The 1848 definition of noma by Tourdes is similar to the modern medical understanding of the disease: "a gangrenous disease affecting the mouth and face of children living in bad hygiene conditions and suffering from debilitating diseases, especially eruptive fever, beginning with an ulcer on the oral mucosa rapidly spreading outside and destroying the soft and hard tissues of the face and almost always fatal" [13].

In Laos, the name commonly used for noma is 'Pagnad Pak Poue' meaning 'disease of mouth rotting' [14]. In Zambia, the disease has been labelled as 'aka popo', meaning the child has been fed a stillborn fetus, and the flesh is 'coming out' (describing the sloughing of the cheek) [15]. In Hausa, the most widely spoken language in northwest Nigeria, several names for noma have been documented including 'ciwon iska', 'bakin kare' [5], 'danhurawa', 'tuareg', 'akin' [16], 'gaude' and 'sadde' [17]. Several of these names are generalised terms and have reportedly caused confusion in patient recruitment drives for surgical interventions, as patients with ailments such as cleft lip and palate also identify with these names[5]. A further complicating etymological factor in the setting is that the word 'noma' means 'farming' in Hausa [16]. Names form a part of the understanding of the disease, and in this case, the beliefs about the causes of noma such as spirits, living creatures (insects and animals), and connections with previous illness [16]. These names and the beliefs about the disease have an impact on processes such as health-seeking behaviours and stigmatisation. If it is believed that the disease is caused by spirits or a bad omen, patients and their families are more likely to be ostracised [18].

## Epidemiology

There is a shift in reporting of noma from primarily in Europe and India in the 1800's [4,10,12,19–32], to parts of Africa and North America in the 1900's [11,15,17,22,33–77], to Africa, South America and Asia in the 2000's [6–9,14,16,78–150] (Fig 2). Noma cases were reported in Irish and British soldiers in India the 1880's [26,28]; in Belsen and Auschwitz concentration camps during the Second World War [9,14,17,45,139,151,152] and in the general war-time population of the Netherlands following the famine in the winter of 1944/1945 [139]. Since the Second World War, as living conditions improved, the occurrence of noma in Europe dramatically decreased and is only sporadically reported in the region today [80,81,83,129]. In recent years, noma has been reported in many countries around the world, but primarily in low and middle income countries in Africa and Asia (Fig 2).

In 2007, the WHO carried out a survey in African member states, which found that 39 of the 46 countries surveyed had reported noma cases in the year prior to data collection [153]. Those with the highest number of reported cases were Burkina Faso, Ethiopia, Mali, Niger,

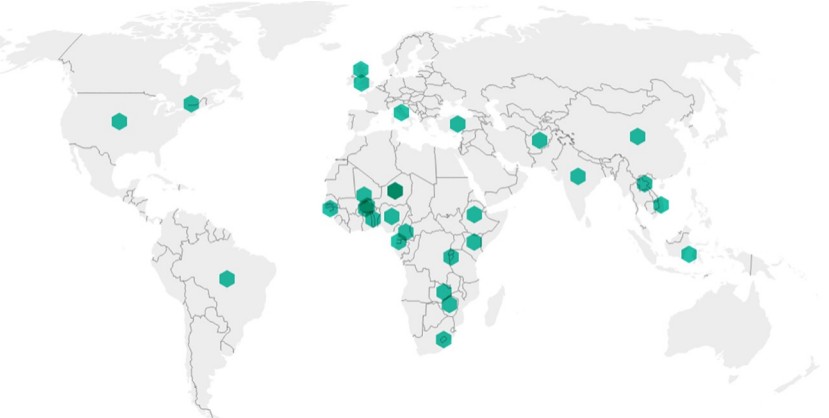

Created with Datawrapper

**Fig 2.** *Map of location of noma studies published from 2000 to 2021 included in this review* [6–9,14,16,78–150] *(green dot represents at least one study in that country) (Created using Datawrapper, basemap*: https:// datawrapper.dwcdn.net/RE1zh/1/).

Nigeria and Senegal which led to these countries being labelled the 'noma-belt' [153]. This term is commonly used when reporting the epidemiology of noma. However; the information gathered that led to this term was not standardized and not based on robust global prevalence or incidence estimates across countries [153]. Since 2000, cases have also been reported in a wide range of settings (Fig 2), indicating a much wider distribution than the usually reported 'noma-belt' [153].

The oldest estimate we found of the burden of disease, based on hospital admissions, indicated that, noma was diagnosed once out of every 5,000 cases of children admitted to hospital with an illness, between 1860 to 1871 in Edinburgh [4]. In 1997, Barnes *et al.* estimated that, based on records from three referral centres, the prevalence of noma was 1 case per 1,250 children aged two to six years per year in Nigeria [154]. In 1998, the World Health Organisation (WHO) estimated that 140,000 new cases of noma occur each year globally and that 770,000 patients were living with noma sequelae at that time [155], the origin of this estimate is unclear [156]. In 1999, it was estimated that there was an annual incidence of 4.2 acute noma cases per million Senegalese children aged 1–4 years [157]. This estimate was calculated using a WHO recommended formula (S1 Equation), based on a 5–20% presentation rate of patients with acute noma or sequelae, and an 80–90% mortality rate in the acute stages of the disease [157]. A Nigerian study (retrospective chart review from 2010 to 2018) estimated the incidence of noma in the north central zone was 8.3 per 100,000 population members at risk [135] and a further study from Nigeria (2018) estimated the community-based point prevalence in the northwest was 3,300 out of every 100,000 children aged 0–15 years [142]. The large variation in these results is due to the differing study designs and the different stages of noma (and case definitions for these stages) included in the estimates.

Through a retrospective chart review (n = 6,390) in 2003, Denloye *et al.* estimated seven cases per 1,000 children aged between one and 16 years had noma between 1986 and 2000 in Nigeria [118]. In that same year, a Fieger *et al.* modelled the incidence of noma in northwest Nigeria based on the number of clefts and concluded that the incidence of noma is estimated to be 6.4 per 1,000 children from 1996 to 2001 [139]. These estimates may not accurately reflect the present incidence of acute noma or the prevalence of patients with noma sequelae as they

are based on expert opinion or historical data. It is also unclear which stages of noma are included in these estimates [158].

## Risk factors

There is limited primary evidence on the risk factors for the development of noma. The table below explores the risk factors noted in the primary studies included in this review (Table 1). Reported risk factors for the development of noma in these primary studies include chronic malnutrition [11,15,17,75,111,114,118,147], comorbidities either at the time of noma diagnosis or in the three months leading up to diagnosis [11,15,17,41,51,75,104,107,111,114,118,122,147] and low vitamin A and vitamin C levels [7]. Social and environmental risk factors include being between two and five years of age [11,15,41,51,75,96,107,118], not being breastfed [114,143], lack of access to basic health care [41]- including a lack of childhood vaccinations [100,143], poor oral hygiene practices leading to gingivitis (Stage 0 noma) [100], low socioeconomic status [104], a lack of variety in the diet [143], the mother being unmarried, not the primary caretaker [143], and having a high number of previous pregnancies [111], and the absence of chickens at home [111].

Other studies have hypothesized further risk factors for noma development including household variables, such as proximity of livestock to living areas and poor sanitation [100], which is thought to lead to possible contamination of water and food sources and consequently increasing the risk of infections [159,160]. However; caution is needed when interpreting these findings as they are based on the proportions of cases vs controls having these risk factors and more robust evidence is needed to validate these findings.

Reported comorbidities in the primary studies (case control, cohort, retrospective chart reviews) include malnutrition [11,75,107,111,114,118], respiratory disease [104,111], diarrhoea [11,111], HIV [96,104,122], malaria [104,118] and vaccine preventable diseases, specifically measles [11,107,118]. Most of the case reports and case series (n = 68) list at least one comorbidity (103 comorbidities listed in these case reports and case series). The most widely reported comorbidities in the case reports and case series included in this review are malnutrition [14,22,26,36,37,44,56,78,83,100,103,106], HIV [56,60,82,87,90,150], anaemia [8,96,106,112] and measles [4,39,49,161] (Fig 3). As this information is based solely on case reports and case series, primarily reported from health care centres, no causal link or strength of association can be measured. Infections are usually the product of a compromised host and a single offending agent or multiple offending agents. Due to challenges with conducting scientifically robust risk factor analysis for noma, it is difficult to separate comorbidities from predisposing conditions and true causative factors.

One theory for the higher incidence of noma in children aged two to five years, is that this is the teething age when deciduous teeth are formed [41]. This formation slows down the circulatory flow to the gums due to compression, leaving the oral cavity more susceptible to infections [41]. A Zambian study postulated that during the weaning period from breastfeeding, children eat more solid food, which was less nutritious and less sterile than breast milk, and this placed them at potential risk for noma development [15]. Another study showed that if weaning foods are prepared under unhygienic conditions, they are frequently contaminated with pathogens and are a major factor in the cause of diarrhoeal diseases [162], a further reported risk factor for noma [163].

Studies that attempted to identify risk factors for noma were hampered by the retrospective nature of case ascertainment limiting the kinds and standardization of risk factor data collected [11,15,75,96,104,107,118,122]. The absence of a suitable control group precluded the ability to find associations between noma and potential risk factors in some studies [11,15,41,96,164]. In other studies there was no statistically robust examination of risk factors

**Table 1. Risk factors for noma identified in primary research.**

| Study | Study details | Risk factors identified |
|---|---|---|
| Osuji, 1990 [51] | Study type: Cross-sectional<br>Location: Nigeria<br>n: 58 cases of acute necrotizing gingivitis (Stage 1 noma as categorized under the WHO system [1]), 5 noma cases (diagnosed as advanced acute necrotizing gingivitis with sequestrum formation) | • Respondents aged between 2–7 years (n = 49, 85% acute necrotizing gingivitis cases, n = 3, 60% noma cases)<br>• Rainy season (n = 42, 67%)<br>• History of recent febrile illness (n = 55, 87%) |
| Lazarus, 1997 [11] | Study type: Retrospective chart review, reviewing charts of cancrum oris patients from the previous 35 years<br>Location: South Africa<br>n: 26 respondents | • Respondents mean age 4 years 4 months (range 1–15 years)<br>• Malnutrition (n = 7/ 11 (whose records had comorbidity information), 64%)<br>• Gastroenteritis (n = 4/11, 36%)<br>• Measles (n = 3/11, 27%) |
| Nath, 1998 [15] | Study type: Retrospective chart review over 15 years<br>n: 81 respondents | • Respondents aged between 1 and 4 years (n = 67, 83%),<br>• Diarrhoea (n = 13, 28.9%)<br>• HIV (n = 26/45 (children admitted between 1989–93), 60.5%),<br>• Malnutrition (n = 15/ 45 (no. children assessed for malnutrition), 33.3%)<br>• Rainy season (n = 60, 74.1%), |
| Ndiaye, 1999 [41] | Study type: Prospective cohort<br>Location: Senegal<br>n: 25 later stage noma cases, 1058 acute necrotizing gingivitis cases | • Noma respondents mostly aged >15 years (n = 13, 52%), acute necrotizing gingivitis respondents mostly aged between 1–4 years (n = 465, 44%)<br>• No access to basic care (n = 20/25 (noma only), 80%) |
| Enwonwu, 1999 [17] | Study type: Case control study<br>Location: Nigeria<br>n: 86 noma cases | • Respondents mean age 5.9 years (Standard Deviation) (SD) 2.6 years<br>• Malnutrition (Weight-for-height Z score (WHZ) $\leq$ -2.0 SD) (n = 9, 10.2% controls, n = 17, 19.4% cases) |
| Oginni, 1999 [75] | Study type: Retrospective chart review of noma patients from 1982 to 1996<br>Location: Nigeria<br>n: 146 noma patients, 133 acute, 13 sequelae (which was 1.7% of all patients admitted to the hospital during this time). | • Respondents mean age 4.7 years (SD 2.6 years)<br>• Malnutrition (n = 146, 100%)<br>• Poor oral hygiene (n = 122, 83.6%) |
| Denloye, 2003 [118] | Study type: Retrospective chart review 1986 to 2000<br>Location: Nigeria<br>n: 45 noma cases | • Respondents mean age 4.2 years (SD 2.7 years)<br>• Malnutrition (n = 45, 100%)<br>• Malaria (n = 14, 31%)<br>• Measles (n = 14, 31%) |
| Enwonwu, 2005 [114] | Study type: Case control<br>Location: Nigeria<br>n: 91 noma cases | • Respondents mean age 2.6 years (SD 1.0)<br>• Malnutrition (median height for age z-score noma group -3.74, control group 1–1.41, control group 2 0.85) |
| Phillips, 2005 [7] | Study type: Case control<br>Location: Nigeria<br>n: 68 noma acute cases | • Biological markers suggestive of malnutrition (lower plasma levels of vitamin A (p<0.001), vitamin C (p<0.05) and zinc (p<0.001))<br>• Marked reductions (p<0.001) in albumin and blood haemoglobin |
| Chidzonga, 2008 [96] | Study type: Retrospective chart review of charts between 2002 and 2006<br>Location: Zimbabwe<br>n: 48 acute noma cases, all HIV positive (by design) | • Respondents aged <16 years (n = 11, 64.7%)<br>• Gender (female n = 31, 64.6%) |
| Millogo, 2012 [122] | Study type: Retrospective chart review from 1988 to 2007<br>Location: Burkina Faso<br>n: 212 patients (n = 14, 6.6% had HIV) | • Respondents mean age 15.3 years for HIV group, 4.7 years for non-HIV infected group<br>• Concurrently HIV-infected patients had higher mortality (38% vs 6%) |
| Baratti-Mayer, 2013 [111] | Study type: Case control<br>Location: Niger<br>n: 82 cases and 327 controls | • Respondents aged 0–12 years<br>• Severe stunting (Height-for-age Z score $\leq$3 SD) (Odds Ratio) (OR) 4.87, 95% Confidence Interval (CI) 2.35–10.09)<br>• Wasting (WHZ $\leq$3 SD) (OR 2.45, CI 1.25–4.83),<br>• High number of previous pregnancies in the mother (OR 1.16, CI 1.04–1.31)<br>• Presence of respiratory disease, diarrhoea or fever in the 3 months prior to data collection (OR 2.70, CI 1.35–5.40)<br>• Absence of chickens at home (OR 1.90, CI 0.93–3.88) |

(*Continued*)

**Table 1.** (Continued)

| Study | Study details | Risk factors identified |
|---|---|---|
| Konsem, 2014 [104] | Study type: Chart review 2003 to 2012<br>Location: Burkina Faso<br>n: 55 acute noma cases | • Respondents mean age 7.64 years<br>• Concomitant Bronchopneumonitis (n = 20, 36.4%)<br>• Malaria (n = 14, 25.4%)<br>• HIV (n = 11, 20.0%)<br>• Low standard of living (n = 21, 38.2%)<br>• Anaemia (n = 14, 25.4%) |
| Braimah, 2017 [147] | Study type: Retrospective chart review from 1999 to 2011<br>Location: Nigeria<br>n: 159 acute noma cases | • Mean age was 3.34 ± 2.2.<br>• Measles 75 (47.2%), followed by<br>• Protein-energy-malnutrition 67 (42.1%). |
| Adeniyi, 2019 [107] | Study type: Retrospective chart review from 1999 to 2011<br>Location: Nigeria<br>n: 159 acute noma cases | • Respondents aged between 1–5 years (n = 139, 87.4%)<br>• Concurrent disease at presentation or in the 3 months preceding their presentation at the hospital (n = 148, 93.1%)<br>• Measles (n = 75, 47.2%)<br>• Protein-energy malnutrition (n = 67, 42.1%) |
| Farley, 2019 [143] | Study type: Case control study<br>Location: Nigeria<br>n: 74 cases and 222 controls | • Respondents median age 5 (IQR 3, 15)<br>• Vaccination coverage documented on vaccination cards for polio and measles was below 7% in both groups<br>• Child being fed pap every day (OR 9.8; CI 1.5, 62.7);<br>• Potential protective factors including:<br> ○ the mother being the primary caretaker (OR 0.08; CI 0.01, 0.5)<br> ○ the caretaker being married (OR 0.006; CI 0.0006, 0.5)<br> ○ colostrum being given to the baby (OR 0.4; CI 0.09, 2.09) |

WHZ- weight-for-height Z score; OR = Odds Ratio; SD = Standard Deviation; CI = Confidence Interval; WHO = World Health Organisation; HIV = human immunodeficiency virus

using proven statistical methods such as multivariable regression (which would adjust for confounders), limiting the validity and reliability of results [17,41,51,75].

## Aetiology

The pathogenesis of noma is poorly understood. Strikingly, this quote from a paper written in 1893 still partly reflects the current debated nature of the pathogenesis of noma "There must

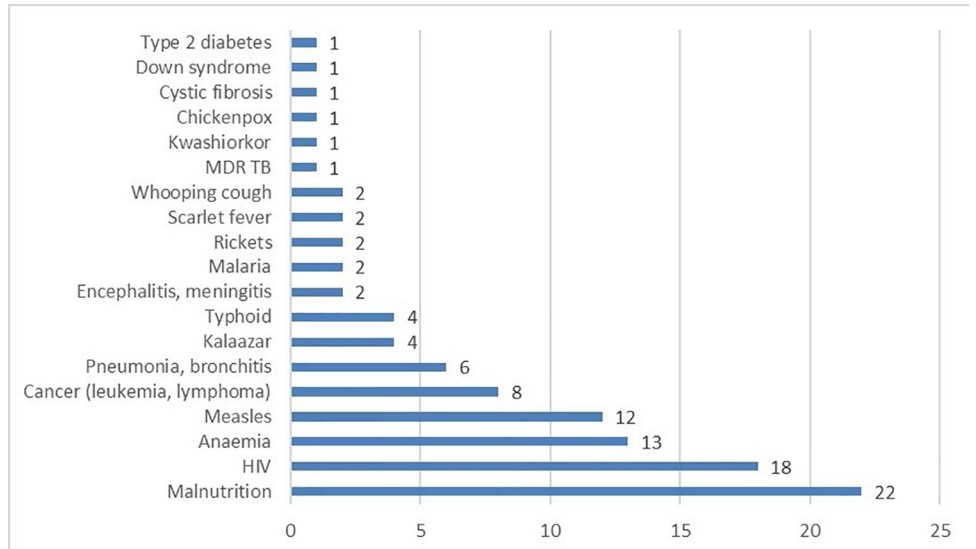

**Fig 3. Comorbidities associated with noma in case reports and case series (N = 103).**

surely be a specific organism and a combination of predisposing causes, not poverty alone, but poverty plus a sickly habit of body" [12].

A range of organisms have been identified in the oral flora of noma patients, but none have been consistently present, casting doubt on a specific organism's role in the development of noma [7,17,38,40,98,111,165]. Other studies have noted that the characteristics of noma are similar to that of an opportunistic infection, implicating a change in the equilibrium of commensal bacteria due to a derailment of host defences [11,96,98,166]. Evidence that supports the understanding of noma being an opportunistic infection rests in the fact that most cases have concurrent infections or occur in immunocompromised individuals [11,15,17,41,51,75,104,107,111,114,118,122,147]. Table 2 below offers a summary of the etiologic studies included in this review and the organisms identified, the details of each study and limitations of the study methods.

## Clinical progression

While the clinical manifestations and sequelae of noma in each case are unique, the infection invariably starts with inflammation of the gums, which then leads to ulceration and the rapid destruction (within weeks [1]) of the cheek and in some cases the jaw, lip, nose and/or the eye [15,148,167]. For the purposes of case detection, the World Health Organisation has classified noma into stages, the first stages (Stage 0 to 4) are the acute stages of noma lasting only a few weeks; Stage 0: simple gingivitis; Stage 1: acute necrotizing gingivitis; Stage 2: oedema; Stage 3: gangrene. Stage 4: scarring and Stage 5: sequelae [1]. Deaths in noma patients are primarily reported to be due to starvation, aspiration pneumonia, respiratory insufficiency or sepsis [13,128]. Even though noma primarily affects young children [2], noma cases in adults, mostly in conjunction with other severe infections (like HIV, cancer or oral myiasis) have been reported [76,82,95,150].

Treatment with antibiotics, wound debridement, and nutritional support in the early, reversible stages of the disease can reduce the duration and severity of the acute phase of noma and the extent of tissue damage, thus reducing mortality and morbidity of noma (discussed in detail below) [2]. Those who survive the acute stages will often have severe sequelae including difficulty eating, seeing and breathing [1,2,136]. Survivors often need complex surgical reconstruction to restore form and function [167]. Trismus (a restriction in mouth opening) is one of the most disabling sequelae [91,144] and can lead to complications such as aspiration, malnutrition, poor oral hygiene, speech deficits, a compromised airway, and pain [168].

The aesthetic and functional sequelae of noma are compounded by the psychological impacts of the disease, not only on the patient, but also on family members and caretakers. Studies have reported that noma has led to mental health issues due to the social isolation and shunning of survivors and their families, bullying, a lack of access to education, difficulties finding jobs, and limited marital prospects [16,102,141,156,163,169–172].

There have been no published reports of noma re-activating [163]. None of the literature included in this review provided evidence to suggest that noma was contagious [163]. Given the ethical and practical challenges of conducting studies to assess the progression of the disease among the hard-to-reach affected communities, information is lacking on the environmental, nutritional and physiological conditions that trigger the progression to the later gangrenous stages of the disease.

## Treatment

**Acute phase treatment (Stages 0 to 4).**   Historically (1800's and early 1900's), noma cases presenting at medical institutions were managed with nutritional support with high protein

**Table 2. Microorganisms found in the oral flora of noma patients by year.**

| Study | Study details | Organism | Limitations |
|---|---|---|---|
| Falkler, 1999 [38] | Study type: Cross-sectional study<br>Location: Nigeria<br>n: Eight cases<br>Additional details: Cancrum oris lesions (present for six weeks to two years) were cultured for anaerobic microorganisms. | *Fusobacterium necrophorum* and *Prevotella intermedia* were isolated from seven and six of the eight lesions, respectively. | Long duration of infection before testing (up to two years), small sample size, no healthy matched comparison group. |
| Phillips, 2005 [7] | Study type: Case control study<br>Location: Nigeria<br>n: 68 acute noma cases, 63 village and 45 urban controls<br>Additional details: Cases were found over four years through house visits. Controls were matched by age and were children attending out-patient clinics and primary health care centres for routine checks, and had no recent history of any disease, fever and diarrhoea. Oral bacteria were studied by polymerase chain reaction on six cases. Excluded those treated with antibiotics or traditional medicine in last 48hrs. Excluded measles, HIV and malaria comorbid patients. | Bacteria observed at the highest frequencies in noma lesions were *Prevotella intermedia* (83%), *Tannerella for sythensis* (83%), *Porphyromonas gingivalis* (50%), *Campylobacter rectus* (50%) and *Treponema denticola* (50%). | Control selection (children attending health care facility) could have biased results as these children were already accessing care. It is unknown how long each patient had noma for. The sample size for bacterial testing was small (n = 6). |
| Chidzonga, 2008 [96] | Study type: Retrospective chart review<br>Location: Zimbabwe<br>n: 48 acute noma cases, five cases had microbiologic investigations<br>Additional details: All cases presented one to two weeks after onset of symptoms | *Staphylococcus aureus, Klebsiella species, group D Streptococcus, and group B hemolytic Streptococcus.* | Small sample size, retrospective chart review, no control group. |
| Baratti-Mayer, 2013 [111] | Study type: Prospective matched, case-control study<br>Location: Niger<br>n: 82 acute noma cases, 327 matched controls<br>Additional details: Study took place over six years. Exact stage of noma cases not defined. Controls matched on age and home village. Extracted total genomic deoxyribonucleic acid. Cases who received antibiotics or whose specimens deteriorated were excluded (n = 20), 117 microbial samples were processed from noma cases and 235 from controls. Multivariable model showed organisms associated with noma. | A reduced proportion of *Spirochaeta, Fusobacterium, Capnocytophaga,* and *Neisseria* in the oral microbiota, but an increased proportion of *Prevotella* associated with noma. Controls had higher *Fusobacterium* genus levels raising doubts about previous findings. | Controls were significantly older than cases. 28% of observations in the analysis were excluded because of missing data for microbiological variables due to problems collecting data due to poor health. |
| Huyghe, 2013 [93] | Study type: Case control study<br>Location: Niger<br>n: 84 acute noma cases, 37 acute necrotizing gingivitis cases and 343 controls<br>Additional details: Cases had no antibiotics, no dental cleaning and did not receive fortified food during the 3 previous months. Subjects with lesions older than 4 weeks were excluded. | Compared to the healthy controls, a lower bacterial diversity was found in noma samples. Less *Porphyromonadaceae, Tannerella spp., Capnocytophaga spp., Fusobacteria* and *Cetobacterium spp.* were found in noma samples. Raises doubts about *Fusobacterium necrophorum.* | Authors state need for time series data and the utilization of high-throughput sequencing capacity to elucidate the aetiology of noma. |

foods (fruit, eggs, milk, meat [12,28,37]) alcohol (wine, brandy and whisky [10,12,28,31,32]) and wound cleaning using bicarbonate of soda [27,28], leeches, [27] and nitric acid [10,26,161]. It is difficult to know whether these methods were beneficial as all evidence was derived from case series and case reports. However, these treatments clearly point to an early appreciation of poor nutrition and hygiene being contributing factors to the progression of the disease.

More recently (later 1900's and 2000's), timely administration of broad-spectrum antibiotics [14,85,92,96,97], wound cleaning and debridement [9,17,73,78,84,97,106], and nutritional support [13,92,96,173] have shown to be effective in reducing the severity and sequelae of noma by arresting the acute phase of the infection in some patients. A range of antibiotics were reported in the included studies such as amoxicillin [78,84,97,106], metronidazole [8,9,38,84,103], lincomycin [80] and cefotaxime [8,81]. No studies comparing the relative efficacy of these antibiotics were identified.

The current WHO guidelines for the management of the acute stages of noma in clinical settings includes [1]: oral hygiene (mouth wash Chlorhexidine 0,2%, 10 ml), antibiotic treatment (amoxicillin and metronidazole), nutritional support (high protein), wound cleaning (compresses soaked in hydrogen peroxide) and dressing (honey for local dressing and for antibacterial action and regeneration) [1].

**Sequelae treatment (Stage 5).**   If the patient survives the acute illness, they can live into adulthood but often require extensive reconstructive surgery and intensive physiotherapy to improve the resulting structural and functional defects [174] that often require a number of surgical treatments [137]. Studies have highlighted the fact that the time between acute illness and surgical care can be decades [136,141,144]. The clinical manifestation of each noma case is unique, and as such, the surgical procedures used to treat each noma case differ [15,39,42,74,102,115–117,145,148]. Reported surgical techniques include pedicled supraclavicular flaps for the treatment of large unilateral facial defects [102,117]; myocutaneous submental artery flaps, bony and/or soft tissue trismus releases [109], forehead flaps [109,144] and lower lid ectropion release [109]. In one study, extra-articular ankylosis due to noma was treated using soft tissue reconstruction with large free flaps [116]. Trismus was treated using a bone distractor in one study [134], and in another mouth opening was performed by bone-bridge excision, sometimes associated with contralateral coronoidectomy [116]. In a further study, the reconstruction of an upper lip defect was conducted using Gillies fan flaps [89]. A 2006 book on noma surgical techniques includes information on the reconstruction of the lips and corner of the mouth using Abbe, Estlander and fan flaps; and the reconstruction of the cheek using temporo-parietal fascia and deltopectoral flaps; and the reconstruction of central defects using radial forearm and local turnover flaps [167]. Challenges with anaesthesia for noma survivors have been reported, particularly in patients with severe trismus [125,127].

Physiotherapy is an essential part of noma treatment, especially to prevent or minimise trismus [74] and can lead to improvements in eating, chewing and speaking [112].

Noma often leads to stigmatization and resultant social isolation of the patients and their family members from their communities [2,14,16,102,156]. Several studies have highlighted the importance of including social and psychological support for noma patients and their families [136,137,156,169].

Outcomes of noma treatment are difficult to ascertain due to inconsistent patient follow-up in most studies [91]. This is mostly due to the remote locations of the home villages of patients and difficulties in accessing health care facilities [13,14,113]. However, there have been evaluations of noma survivors after surgery which have shown that surgical treatment for noma survivors greatly improves their quality of life, even if functional improvements (specifically mouth opening) are not pronounced [14,102,136,141,156].The extent of long-term sequelae and their impact on quality of life of noma patients depends on the severity of the disease at initial presentation, efficacy of antibiotic treatment, wound debridement and the facial structures affected [91,92,96,97]. It was noted that a validated, standardized noma patient-reported outcome measurement tool would be helpful in standardizing outcome reporting after surgical treatment [136,141,175].

**Traditional treatments.**   In Mali and Nigeria, traditional healers' knowledge of noma was limited [126,146], however, several traditional healers in Nigeria reported treating different stages of the disease. In Nigeria, traditional treatments for noma include ground herbs, plants, ointments and piercing of the swollen cheek (in the oedema phase of noma) [126]. Traditional healers in Nigeria reported referring patients with the later stages of disease to (hospitals and clinics and being interested in assisting with referrals of noma patients, and attending trainings on the disease [126].

## Mortality

There is limited and inconclusive evidence around the pathogenesis of noma leading to death. Factors that favour survival (apart from antibiotic treatment and wound debridement) are unknown. The mortality rate of noma depends on multiple factors and is poorly enumerated. The WHO (based on expert opinion and retrospective chart analyses) states that noma has a mortality rate of 90% within weeks after the onset of noma if left untreated [1]. The speed with which death occurs is also debated, some state death occurs in as little as two weeks from the onset of first symptoms [1] but it is unclear which symptoms these are. The clearest reported estimate is that death can occur in a matter of days after the onset of oedema [13]. What is certain is that when noma is identified and treated in a timely manner, mortality greatly decreases [176].

Table 3 explores the mortality rates reported in various studies included in this review. These estimates highlight the differences in mortality rates in groups who received no antibiotic treatment (49–94%), compared to those who had received drug therapies such as antibiotics (0–38%) (Table 3). It should be noted that these estimates are derived from case series and

**Table 3. Mortality reported in included studies.**

| Study | Location | Study design | Cases | Mortality (%) | Treatment |
|---|---|---|---|---|---|
| Tourdes, 1848 [177] | Europe | Case series | 239 | 73% | No drug therapy |
| Barthez, 1855 [177] | Europe | Case series | 29 | 89% | No drug therapy |
| Ritchie, 1872 [4] | Europe | Case series | 8 | 63% | Iron with citric-acid, nutritional support |
| Springer, 1904 [177] | Europe | Case series | 88 | 94% | Wound debridement |
| Gupta, 1945 [44] | India | Case series | 79 | 49% | Pentavalent antimony (treatment of leishmaniasis), nutritional support, vitamins, blood transfusions, local antiseptic treatment |
| Jelliffe, 1952 [37] | Nigeria | Case series | 53 | 30% | Penicillin |
| Mehrotra, 1966 [49] | India | Case series | 20 | 15% | Antibiotics, multivitamins, high protein diet, sequestrectomy, plastic reconstructive surgeries |
| Adekeye and Ord, 1978–1982 [13] | Nigeria | Case series | 13 | 0% | Antibiotics |
| Bourgeois, 1981–93 [13] | Senegal | Case series | 73 | 10% | Drug therapy, kind of treatment not specified |
| Oginni, 1982–96 [13] | Nigeria | Case series | 133 | 0% | Drug therapy, kind of treatment not specified |
| Nath, 1998 [15] | Zambia | Retrospective chart review | 117 | 20% | Nutrition, wound care |
| Chidzonga, 1996 [48] | Zimbabwe | Case series | 8 | 38% | Antibiotics, wound debridement, removed mobile teeth, irrigated wounds |
| Millogo, 2012 [122] | Burkina Faso | Retrospective chart review | 212 | HIV 38%; non- HIV patients 6% | Antibiotics, anti-retroviral therapy |
| Konsem, 2014 [104] | Burkina Faso | Retrospective chart review | 55 | 15% | Antibiotics |
| Braimah, 2017 [147] | Nigeria | Retrospective chart review | 159 | 25% | Antibiotics |

retrospective chart reviews; no standardized reporting of noma stage was used and the studies do not have the same follow-up periods. The evidence should be evaluated with these study design restrictions in mind, as they could over- or under-estimate the mortality rates of noma patients, particularly at the community level.

## Discussion

There is a dearth of research and literature on noma. The date of the first study included in this review was 1843, and since this time, an average of one publication has been written on the disease per year (calculated based on the studies included in this review). Despite significant progress in scientific methods since the first study, the literature remains predominantly populated with case reports and case series. More scientifically robust studies are needed. The reasons behind this neglect include the lack of knowledge about the disease by healthcare workers, in part due to noma not being included in medical curricula leading to under-reporting and misdiagnosis of cases [92,140], the hypothesized low prevalence of the disease [1], which may, in part, be due to inconsistent surveillance and reporting on the disease [107], the relative inaccessibility of the affected communities and the rapid progression of the disease, high mortality, stigmatization and isolation of noma survivors [2,16,102,156,176]. There is appreciation among the small community of clinicians and researchers involved in noma care and research that this lack of awareness impacts on the ability to develop and implement sound evidence-based policies and public health initiatives aimed at eradicating noma from communities that continue to be afflicted by this ancient disease. Several studies have stated that these gaps in research could be filled with better awareness about the disease and call for the inclusion of noma in the WHO list of neglected tropical diseases which would highlight noma in the global health arena [178–180]. It is likely that addressing the causes and conditions contributing to noma will lead to wide ranging benefits.

Based on this literature review, some of the main gaps in knowledge are enumerating the burden of disease (both incidence and prevalence); describing the true mortality rate and pathogenic cause(s) of noma and the role of different comorbidities (specifically measles and HIV) play the development of noma, a finding similar to other reviews [2,13,154,165,174,180–182]. Factors that influence prognosis and the long-term outcomes after care (surgical and non-surgical) [88], including the most effective antibiotic treatment protocols [91], need to be assessed. The knowledge of health care workers about noma in high risk areas, the number of medical school and tropical medicine curriculums that include noma; and the role the varying healthcare actors could play in prevention [183] need to be explored. An additional area for future studies would be to compare prevention methods and messaging [92] to identify the most effective mechanisms. Efforts to eliminate extreme poverty may lead to a reduction in the number of cases of noma, and potentially even eliminate this disease.

There were several limitations to this review. Given the inclusion time period of this review (from the 1800's to present) it is likely that some manuscripts (especially in the earlier years) were not available on current indexing systems and hence not included in this review. We used Google Translate to translate non-English papers, which could have led to some misinterpretation, as it is not an official academic translating service. The inclusion of published manuscripts only and not books and other grey literature could have limited the amount of information identified.

In summary, noma is a preventable disease that affects young children in the most vulnerable and impoverished communities. It is a devastating and often fatal disease that requires urgent and intensive clinical and surgical care, often difficult to access as most cases of noma occur in resource-limited settings. Noma has been reported in the literature for hundreds of

years; however major gaps in knowledge about the disease still exist. What is clear from the literature is the wide geographical spread of noma, and the need for further studies to gain an understanding of the burden and distribution of disease; the true mortality rate, and the pathogenic cause(s) and the factors that influence prognosis and outcomes after treatment. Filling these gaps in knowledge will help with the development of effective targeted interventions to reduce the burden of noma in the most affected populations.

## Supporting information

**S1 PRISMA Checklist. Preferred Reporting Items for Systemic review and Meta-Analyses (PRISMA) requirements.**
(DOCX)

**S1 Data. Full list of included articles.**
(DOCX)

**S1 Equation. WHO formula.**
(DOCX)

## Acknowledgments

The authors are grateful to all of the all of the authors whose work is cited in this review. A big thank you to Wendy Smith for helping to source the articles through the University of Cape Town medical library.

## Author Contributions

**Conceptualization:** Elise Farley, Ushma Mehta, Annick Lenglet.

**Data curation:** Elise Farley.

**Formal analysis:** Elise Farley.

**Methodology:** Elise Farley.

**Supervision:** Ushma Mehta, M. Leila Srour, Annick Lenglet.

**Validation:** Ushma Mehta, M. Leila Srour, Annick Lenglet.

**Visualization:** Elise Farley.

**Writing – original draft:** Elise Farley.

**Writing – review & editing:** Ushma Mehta, M. Leila Srour, Annick Lenglet.

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
