## [Decision Letter · Decision Letter 0]

13 Oct 2021

Dear Dr Farley,

Thank you very much for submitting your manuscript "Noma (cancrum oris): a scoping literature review of a neglected disease (1843 to 2021)" for consideration at PLOS Neglected Tropical Diseases. As with all papers reviewed by the journal, your manuscript was reviewed by members of the editorial board and by several independent reviewers. The reviewers appreciated the attention to an important topic. Based on the reviews, we are likely to accept this manuscript for publication, providing that you modify the manuscript according to the review recommendations. 

Sincerely,

Joseph M. Vinetz

Deputy Editor

Joseph Vinetz

Deputy Editor

Reviewer's Responses to Questions

**Key Review Criteria Required for Acceptance?**

**Methods**

-Are the objectives of the study clearly articulated with a clear testable hypothesis stated?

-Is the study design appropriate to address the stated objectives?

-Is the population clearly described and appropriate for the hypothesis being tested?

-Is the sample size sufficient to ensure adequate power to address the hypothesis being tested?

-Were correct statistical analysis used to support conclusions?

-Are there concerns about ethical or regulatory requirements being met?

Reviewer #1: Yes. The review strategy is clear

Reviewer #2: (No Response)

**Results**

-Does the analysis presented match the analysis plan?

-Are the results clearly and completely presented?

-Are the figures (Tables, Images) of sufficient quality for clarity?

Reviewer #1: In the earlier reports ( lines 161-164) there is a hint that this was seen in adults formerly e.g. in soldiers . Is this correct ?

Line 204 Is it possible to be more precise about the definitions of the earliest signs eg gingivitis and ulceration. How extensive is this before there is oedema of the facial skin or do we know this ?

Activity lines 258-271. In the more recent oral microbiome studies did any analyse the bacterial flora species level, as in other situations, variation in species levels within a single genus can be significant ?

347-350. Can you enlarge on this section of stigma. For instance provide more information on societal attitudes including discrimination against patients.

Reviewer #2: (No Response)

**Conclusions**

-Are the conclusions supported by the data presented?

-Are the limitations of analysis clearly described?

-Do the authors discuss how these data can be helpful to advance our understanding of the topic under study?

-Is public health relevance addressed?

Reviewer #1: For successful intervention it would be important either to devise a preventive strategy or devise a system for recognising children whose oral condition is at risk of progression. Do mother and child services have a role to play? 

 Is it possible to provide a brief list in table form showing items that could be addressed in order to make a difference?

Reviewer #2: (No Response)

**Editorial and Data Presentation Modifications?**

Reviewer #1: NO

Reviewer #2: In the 1st sentence of the abstract, would place a comma after the word ‘disease’ and before ‘afflicting’

Sentence 2 in the abstract states “in these settings” although those settings have yet to be defined – would suggest rewording to “and this care is often difficult to access, as most cases of noma occur in resource-limited settings.”

In the abstract, I would reverse the order and percentages (placing 1900’s before 1800s) – as currently it is 2000s, then 1800s, then 1900s

In abstract would change aetiology to etiology

I would reword this sentence in the abstract, as I would not consider these ‘themes’; 

Perhaps:

These manuscripts often focused on a number of characteristics regarding noma, including: the history and epidemiology of the disease, …. 

In the last paragraph of the abstract I would remove “many” from “many hundreds of years” and simply state “hundreds of years”. Alternatively, you could also (given this is an encompassing review) be more specific and state when it was first reported in the literature 

While mentioning this point – I would suggest removing the word “scoping” from the title of this article 

In the article, insert a comma after disfiguring in line 20

Line 22 – in the body, again would comment: “in these settings”; “these settings” have yet to be defined in the manuscript – would suggest rewording to “and this care is often difficult to access, as most cases of noma occur in resource-limited settings.”

Line 22 – would remove the word scoping 

Lines 33-35 – again would change this to be chronological order – so 2000s, then 1900s, then 1800s, rather than 2000s, then 1800s, then 1900s as is currently written 

Line 39 – again, would remove “many” from hundreds of years in the literature 

Line 47 – would add a comma after the word disfiguring

Line 49- I would remove the 2 words “to sequelae”

Line 53 – remove “many”

Line 58 – place a comma after the word rate 

Line 65 – place a comma after services

Line 66 – would remove the word microbiology so the sentence clause reads …poorly understood and debated…

Line 68 – would change disfigurements to disfigurement, and would also place a comma after disfigurement 

Line 68- 69 currently written as …..multiple physical impairments such as difficulty in eating, drinking, seeing.

Would change to….and other sequelae, including difficulty with eating and drinking, as well as ophthalmologic and other other visual changes, and difficulty with breathing as a result of upper airway sequelae.

Line 69 – would replace disfigurements with sequelae

Lines 70-71

Currently written: This impose great difficulties on both the survivor and their families. Would suggest rewording to:

Therefore, noma is associated with a high degree of morbidity for survivors, and this often has a significant impact on family members given the long-lasting and often permanent sequelae.

Line 73 – would remove the word scoping 

Line 107 – suggest replacing themes of interest with areas of interest

Line 114 – would carry through and suggest placing “themes” of interests with areas 

Line 115 – aetiology would change to etiology

Line 123, insert a comma after 1843

Lines 121-123 – again would reorder so in chronological order as mentioned above 

Line 130

129. The word ‘noma’ is derived from a Greek word which, loosely translated, means ‘to devour’ 

Suggest rewording to :

The word ‘noma’ is derived from Greek, and loosely translates to ‘to devour.’

Line 135 

lesion on the skin or an internal mucous surface of the mouth 

would reword phrase to

(lesions on the skin or on the internal mucosal surface of the mouth)

Line 151 

Change generalised to generalized 

Line 151 – change which to and 

Line 152 – replace calls with names 

Line 162 – insert a comma after 1900’s

160. Line 164 – change to and in the general population - that is inserting an in in the phrase

Line 168 – insert a comma after the word world

Line 212

Suggest rewording include being aged between two and five years 

To being between two and five years of age 

Line 213 – suggest adding a hyphen after the word healthcare

Line 215- suggest removing the words of the family from the phrase 

Line 215-217, would suggest adding commas, before all the “ands” in these sentences

Line 222, insert comma after variables

Line 231 – changed listed to list 

Line 232 change were to are 

Line 232 change spelling of anemia

Line 233-234

This formation slows down the circulatory flow to the gums due to compression, leaving the oral cavity more susceptible to infections [54]. 

Line 245 – you reference the weaning period – the weaning period from breastfeeding – please define if that is the case 

Line 248 – diarrheal - spelling 

Line 254 – would finish the sentence… potential risk factors in some studies 

Line 258 – would change atiology to etiology 

line 259 - The pathogenesis of noma is poorly understood and the microbiology is debated 

would change to The pathogenesis of noma is poorly understood and debated

line 264 – place a comma after patients

line 265 – place a comma after consistently

line 266-267

currently starts with 

Other studies have noted that noma incorporates the characteristics of an opportunistic infection, 

Would suggest rewording this clause to 

Other have noted that the characteristics of noma are similar to that of an opportunistic infection

Line 269

or immunodeficiency states

rephase to

or occur in immunocompromised individuals 

line 269 – what are where are the concurrent infections reference – please comment on this and provide a citation 

line 277 change manifestation to manifestations 

line 277 change to “are unique”

line 278 starts as an inflammation of the gums 

change to starts with inflammation of the gums 

line 28 - lasting only a couple of weeks 

would define this further

lasting only several weeks or 

lasting 1-2 weeks or or lasting 3-4 weeks or other…

line 288 – insert a comma after debridement

line 288 insert a comma after early 

line 291 – would replace disfigurements with deformities 

line 291 – 292 – this is repetitive (the exact same sentence I believe as above) – see my recommendations on change this sentence, and whether it needs to be repeated again 

line 295 – insert a comma after airway

line 295 – 296 suggest rewording sentence to :

The deformities that result from noma can lead to social stigmatization

Line 305 - biomedical 

I am not sure what biomedical institutions are – can you define this or further define 

Line 307 – insert a comma after leeches 

Line 328 defects [102] that need a comprehensive suite of treatments 

Suggest rewording to

Defects that often require a number of surgical treatments 

Line 329 – suggesting removing the word provision 

Line 367 – I see the use of biomedical again – perhaps I would just change all these biomedical to medical all these implies western medicine, rather than herbal treatments as noted 

Line 368…..patients, and themselves attending didactic training sessions on noma.

Line 371 – There is limited evidence around the pathogenesis of noma leading to death 

Reword to 

There is limited and inconclusive data regarding the risk factors that are associated with mortality

Line 373 – suggest changing enumerated to understood 

Line 376 – suggesting removing about from the sentence 

Line 378

Replace timeously with in a timely manner

Line 382 – insert a comma afte (49-94%) 

Line 384

chart reviews, no 

change to 

chart reviews; no 

line 386 – insert a comma after mind 

line 402 – replace our ability with the ability 

line 413

surgical and non-surgical) insert a comma after this phrase 

line 414 – insert a comma after protocols

sentence starting 415

suggesting rewording to :

Healthcare curriculums, both for practicing clinicians as well as trainees, should include noma – with salient information on the pathogenesis, known risk factors, and importance of timely diagnosis and treatment.

Line 418 – insert a comma after messaging 

Line 418 -420

Currently 

Sincere efforts and political will to eliminate extreme poverty could lead to the elimination of this preventable childhood.

Suggesting rewording to:

Efforts to eliminate extreme poverty may lead to a reduction in the number of cases of noma, and potentially even eliminate this disease. 

Line 422 – replace the review with this review

Line 425 – insert a comma after misinterpretation 

Line 429 – add In summary, to start off the first sentence

Line 430, insert a comma after disfiguring

**Summary and General Comments**

Reviewer #1: If there is one message that comes out of this thorough review of noma it is that it has truly been neglected.

Reviewer #2: Thank you for this detailed and throughout review of noma. I have included changes that are mostly syntax in nature.

PLOS authors have the option to publish the peer review history of their article (what does this mean?). If published, this will include your full peer review and any attached files.

Reviewer #1: No

Reviewer #2: No

Figure Files:

Data Requirements:

Reproducibility:

References

---

## [Editor Report · Decision Letter 1]

1 Nov 2021

Dear Dr Farley,

We are pleased to inform you that your manuscript 'Noma (cancrum oris): a scoping literature review of a neglected disease (1843 to 2021)' has been provisionally accepted for publication in PLOS Neglected Tropical Diseases.

Best regards,

Joseph M. Vinetz

Deputy Editor

Joseph Vinetz

Deputy Editor

---

## [Editor Report · Acceptance letter]

23 Nov 2021

Dear Dr Farley,

We are delighted to inform you that your manuscript, "Noma (cancrum oris): a scoping literature review of a neglected disease (1843 to 2021)," has been formally accepted for publication in PLOS Neglected Tropical Diseases.

Best regards,

Shaden Kamhawi

co-Editor-in-Chief

Paul Brindley

co-Editor-in-Chief
